


# Disentangling the chemistry and transport impacts of the Quasi-Biennial Oscillation on stratospheric ozone

Jinbo Xie[1], Qi Tang[1], Michael Prather[2], Jadwiga Richter[3], Shixuan Zhang[4]

[1]Lawrence Livermore National Laboratory, Livermore, CA, USA

[2]Department of Earth System Science, University of California, Irvine, CA, USA

[3]National Center for Atmospheric Research, Boulder, CO, USA

[4]Pacific Northwest National Laboratory, Richmond, CA, USA

Correspondence to: Jinbo Xie (xie7@llnl.gov)

**Abstract**

The quasi-biennial oscillation (QBO) in tropical winds perturbs stratospheric ozone throughout much of the atmosphere via changes in transport of ozone and other trace gases and via temperature changes that alter chemical processes. Here we separate the temperature-driven changes using the Department of Energy's Energy Exascale Earth System Model version 2 (E3SMv2) with linearized stratospheric ozone chemistry. E3SM produces a natural QBO cycle in winds, temperature, and ozone. Our analysis defines climatological QBO patterns of ozone for the period 1979-2020 using both nonlinear principal component analysis and monthly composites centered on QBO phase shift. As a climate model, E3SM cannot predict the timing of the phase shift, but it does match these climatological patterns. We develop an offline version of our stratospheric chemistry module to calculate the steady-state response of ozone to temperature and overhead ozone perturbations, assuming that other chemical families involved in ozone chemistry remain fixed. We find a clear demarcation: ozone perturbations in the upper stratosphere (above 20-hPa) are predicted by the steady-state response of the ozone column to the temperature changes; while those in the lower stratosphere show no temperature response and are presumably driven by circulation changes. These results are important for diagnosing model-model differences in the QBO-ozone responses for climate projections.





## 1. Introduction

The Quasi-Biennial Oscillation (QBO) is the principal mode of dynamical variability in the
tropical stratosphere, with impact on the circulation and greenhouse gases that extends from the
tropical stratosphere into the troposphere. Its effect on ozone – the most important trace gas in
the stratosphere – has been well studied (Reed 1964; Bowman, 1989; Wang et al., 2022). Despite
being a robust research area for decades, assigning the pattern of ozone perturbations over the
QBO cycle to specific processes is not easy due to the simultaneous temperature and transport
changes (Plumb and Bell, 1982) and the photochemical linkages across most all reactive gases.
This study aims to provide a better understanding of what drives ozone variability over the
QBO cycle.  We develop a new index of the QBO phase from a nonlinear principal component
analysis (NLPCA) of the tropical zonal winds that retains the observed asymmetric pattern and
provides a more consistent measure of the phase throughout the cycle, not just when the zonal
winds change sign.  Second, we create phase-based composite diagrams to investigate the
temporal evolution of ozone patterns, both observed and modeled. Our primary modeling tool is
the Department of Energy (DOE) Energy Exascale Earth Model version 2 (E3SMv2, Golaz et
al., 2022) with interactive stratospheric ozone (Linoz v2, McLinden et al., 2000), and secondarily
we examine some QBO experiments from the National Center for Atmospheric Research
(NCAR) Community Earth System Model (CESM).  We find that QBO cycles in ozone can be
attributed to temperature perturbations in the upper stratosphere (above 20-hPa) and mostly to
circulation changes in the lower stratosphere over a wide range of latitudes.  The observational
data and ozone modeling are described in section 2. The NLPCA method is presented in section
3, followed by the description and use of the Linoz off-line chemistry model in section 4. The
results are in section 5. The discussion and conclusion are in section 6.

## 2. The QBO

### 2.1 Overview

The QBO appears prominently as alternating easterly and westerly equatorial winds that
propagate downward from the top (50 km) to the bottom (16 km) of the stratosphere with a
period of about 28 months (Baldwin et al., 2001; Anstey and Shepherd, 2014; Coy et al., 2016).
Associated with this propagation of the alternating equatorial winds, the QBO also modifies the
vertical propagation of planetary waves and creates global changes in the Brewer-Dobson



Circulation (BDC) (Holton and Tan, 1982; Watson and Gray, 2014; Zhang et al., 2020). Through
perturbations to the BDC, the QBO has been identified as an important source of variability in
the overall chemical composition of the tropical stratosphere (Randel et al., 1998; Shuckburgh et
al. 2001; Park et al. 2017), and it reaches into the troposphere through stratosphere-troposphere
exchange (STE) of ozone (Yang and Tung, 1995; Kinnersley and Tung, 1999) and nitrous oxide
(Hamilton, 1989; Ruiz et al., 2021).
**2.2 Ozone impacts**
The QBO affects ozone through coupled transport and chemical processes, limiting our
ability to ascribe the cause of ozone perturbations to specific processes. Baldwin et al. (2001)
suggest that the dynamic impact of the QBO via direct transport of ozone accounts for most of
the ozone variability. The primary mechanism being: maximum westerly winds correspond to
warmer temperatures that result in diabatic cooling that slower tropical ascent of air parcels, with
the opposite sense (more rapid ascent) for easterly winds. Tropical ozone has a steep, inverted
gradient, 0.1 parts per million (ppm = micromol mol$^{-1}$) at 100 hPa peaking to 10 ppm at 10 hPa.
In this region ozone values are below photochemical steady-state with production exceeding
loss, and thus slower ascent rates lead to greater accumulation of ozone, including in total
column ozone (TCO, Reed, 1964). This ozone anomaly is also impacted by vertical shifts in
NOy (total reactive nitrogen reservoir), which photochemically destroys ozone (Chipperfield and
Gray, 1992; Chipperfield et al., 1994, Politowicz and Hitchman, 1997; Jones et al., 1998). The
QBO pattern in ozone reverses phase outside of the core tropics (15°S - 15°N), consistent with
the return arm of the local equatorial QBO circulation (Holton et al., 1989; Gray and Dunkerton,

1990).

**2.3. The QBO modeling initiative**
Tropical stratospheric variability, in particular the QBO, has been poorly represented in
climate models (Butchart et al., 2011; Butchart et al., 2018; Richter et al., 2020). The number of
models with a naturally generated QBO was 0 in the third Coupled Model Intercomparison
Project (CMIP3); it rose to 5 in CMIP5 and to 15 in CMIP6 (Richter et al., 2020). Even when
models naturally produce a QBO-like variability, the amplitude and periods often fail to match
the observed pattern. In the current Chemistry–Climate Model Initiative (CCMI), many of the
CCMs forced a QBO signal by nudging the equatorial zonal wind (Morgenstern et al., 2017).
Nudging of the winds winds is inherently unphysical and produces an anomalous BDC not found



in the free-running versions of the same CCMs (Orbe et al., 2020).  The World Climate Research
Project (WCRP) Atmospheric Processes And their Role in Climate (APARC) started an QBO
initiative (QBOi) in 2015 to improve CCM simulation of tropical variability (Butchart et al.,
2018), and here we build on those experiments.
**2.4. CCM models**

The primary model for this study is E3SMv2.  E3SM's atmospheric component (EAMv2) is

run as a CCM with specified sea surface temperatures (SSTs) and has 72 vertical layers and a
horizontal resolution of about 100 km. Following Richter et al. (2010), EAMv2 employs gravity
wave (GW) parameterizations that include orographic GWs (McFarlane, 1987), convective GWs
(Beres et al., 2004), and GWs generated by frontal systems (Charron and Manzini, 2002).
Tunable parameters in the orographic and frontal GW parameterizations remain the same as in
EAMv1 (Xie et al., 2018; Rasch et al., 2019). The tunable parameters in convective GWs were
explored to produce a more realistic QBO in EAMv2 with a period around 27 months, much
closer to observations (28 months) as compared to 16 months in EAMv1 (Richter et al., 2019).
Nevertheless, the modeled QBO remains very weak in terms of amplitude. Stratospheric ozone
in E3SMv2 is calculated interactively through transport and the chemical Linoz module
(McLinden et al., 2000; Hsu and Prather, 2009) that was updated from the E3SM O3v1 to O3v2
module (Tang et al., 2021). Linoz v2 data tables are used to calculate the 24-hour-average ozone
tendency (i.e., net production minus loss) from an adopted climatological mean state for key
species ($CH_4$, $H_2O$, and NOy, Cly, Bry) and first-order Taylor series expansions about the local
ozone, temperature, and overhead ozone column (see Eq. (3) in Sect. 5.1).  The data tables are
generated for each year assuming key chemical species ($CH_4$, $H_2O$, and NOy, Cly, Bry) follow
monthly zonal-mean climatologies that scale with the slowly varying changes in tropospheric
mean abundance of their source gases (e.g., $N_2O$, CFCs, halons, $CH_4$, tropopause $H_2O$). The
Linoz model produces a reasonable stratospheric ozone climatology, including seasonal and
interannual variability and the Antarctic ozone hole (Tang et al., 2021; Ruiz and Prather, 2022).
The tropospheric chemical package for E3SMv2 (chemUCI) was not used and the lower
boundary for Linoz was set to 30 ppb. Thus, none of the ozone column variability arises from
tropospheric ozone chemistry. E3SMv2 diagnostics on the tendency of tropospheric ozone
enable the geographically resolved stratosphere-troposphere exchange (STE) flux of ozone every
time step (Hsu et al., 2005; Tang et al., 2013).





The secondary model for this study is CESM2 (Emmons et al., 2020), using a modified
version of the community atmosphere model (CAM) with 83 vertical levels (Randall et al., 2023;
Isla et al., 2024), and also run as a CCM with specified sea surface temperatures (SSTs). CAM
uses the finite-volume dynamical core with a nominal 1° horizontal resolution and with physics
from the Whole Atmosphere Community Climate Model version 6 (WACCM6; Gettleman et al.
2019). The parameters for the convective GW momentum transport were tuned especially for
this version to obtain a realistic, naturally generated QBO. The inline ozone calculation is
replaced with a monthly mean 3D ozone climatology specified from a previous WACCM
simulation. This ozone forcing is formed by merging WACCM simulations for historical (1850-
2014) and future period (2015-2100). The ensemble mean of three historical WACCM
simulations is used for the historical period while one future scenario run is used for future
period. As the mean of free-running CCM simulations, this WACCM ozone climatology does
not have any significant QBO-like variability, and thus it cannot trigger a QBO in the CCM
(Butchart et al., 2023).
With these two different types of simulations, one with interactive ozone and one without, we
must limit our analysis with the pair of models to examining the forced dynamical response
(temperature, circulation), but will use the E3SM results to compare the modeled QBO-ozone
response with observations.
**2.5.  Observed ozone and wind**
For ozone, we derive the observed QBO signal from the monthly zonal mean total column
ozone (TCO) using the Multi-Sensor Reanalysis version 2 data (MSRv2, R.J. van der A, et al.
2015). This latitude-by-month dataset initially covers the period 1979-2012 and later extended to
2020. For stratospheric profiles, we use the zonal monthly mean latitude-by-altitude from the
Concentration Monthly Zonal Mean (CMZM) product (Sofieva et al., 2023). This altitude-by-
month profile data covers the period 1985-2020. The vertical levels are converted to pressure
levels inverting the pressure-altitude formula, $z^* = 16 \log_{10}(1000/P)$ km. We compared this
ozone data with the overlapping period from the Microwave Limb Sounder (MLS) data (V5
Level 3: Schwartz et al., 2021) and found only small differences with regard to QBO patterns.
We use data from the ERA5 reanalysis produced by the European Center for Medium-Range
Weather Forecast (ECMWF) Integrated Forecast System (Hersbach et al., 2020). The version we
use has 137 hybrid sigma model levels from the surface to the model top at 0.01 hPa, and the





horizontal resolution is about 31 km. We use monthly ERA5 data (wind, temperature,
geopotential height) for the period 1979–2020 to analyze the QBO-related dynamical changes,
and 6-hourly ERA5 tropical zonal wind (15°N-15°S) to nudge model simulations mentioned
below. We use the 5°S-5°N tropical average zonal wind from ERA5 and simulations to
determinate the QBO phase index. The combined station zonal wind data from Freie University
of Berlin (Naujokat, 1986) for the period of 1979-2020 is also used in the NLPCA analysis (Fig.

1).

**2.6 The QBOi simulations**
We use a set of three experiments from our two models following the protocol for phase-2 of
the QBOi (Butchart et al., 2018; Bushell et al., 2020; Richter et al., 2020):
(1) Exp1-ObsQBO (nudged): the zonal wind (i.e., u) in the tropical stratosphere is

constrained to follow the observed QBO evolution by nudging it toward ERA5 reanalysis

(Hitchcock et al. 2022).  Thus, the stratospheric climate in the tropics is constrained.

(2) Exp1-AMIP (natural): the zonal wind in the tropical stratosphere evolves freely in each

CCM being forced only by SSTs and trace-gas radiative heating; there is no nudging.

The nudging is applied to the zonal wind over the range 8 hPa-to-80 hPa and 15°S-to-15°N
(Supplementary Fig. 1, nudging coefficient shown is for E3SMv2, that for CESM2 is similar).
There is a slight difference in how the models were nudged: E3SMv2 is nudged to the 3-D ERA5
wind field, while CESM2 is nudged to a 2-D zonally-averaged ERA5 wind field.  The nudging
relaxation timescale is 5 days, which is expected to constrain the slowly evolving QBO winds.

# 3. NLPCA analysis of QBO phase

In diagnosing QBO-related changes to the dynamics and chemistry, we need to define the
phases of successive QBOs, at least and align these phases over a 24-month period. Asymmetric
and nonlinear features of the evolution of the QBO phase are found in many studies (Lindzen
and Holton 1968; Holton and Lindzen, 1972; Giorgetta et al., 2002). The most obvious and sharp
synchronization point is when the QBO west phase (QBOw, i.e. prevailing westerlies) transitions
to the east phase (QBOe: prevailing easterlies) at some pressure level in the middle stratosphere
(taken as 10 hPa here) (Naujokat et al., 1986; Pahlavan et al., 2021; Kang et al., 2022). The
QBOe phase is typically longer (e.g., 63%, Bushell et al., 2019), with wind speeds about twice as
strong as that of the QBOw (Naujokat et al.,1986; Kang et al., 2022). The problem with defining
the QBO phase (index) simply as the absolute time difference relative to the synchronization





point (e.g., Ruiz et al., 2021) is that the duration of different phases varies across successive
QBOs.

Previous use of PCA-derived QBO indices (Hamilton and Hsieh, 2002; Lu et al., 2009) did

not allow for this asymmetric and nonlinear behavior.  Lu et al. (2009) noted that the
reconstructed wind series from the PCA looked more sinusoidal in time than the actual winds,
and thus the asymmetries between phases did not show up in the PCA-based indices. To address
these issues, we use an NLPCA method that utilizes hierarchical-type neural network with an
auto-associative architecture (Scholz et al. 2002). It is a nonlinear generalization of the standard
PCA from straight lines to curves in the original data space, and natural extension to the PCA
method by enforcing the nonlinear components to the same hierarchical order as in the standard
PCA (Scholz et al., 2002). The NLPCA model described here has 5 layers with 3 hidden layers
of neurons. The layers of the neural-network for NLPCA are in the sequence of input-encoding-
bottleneck-decoding-output with the structure of n-(2k+2)-k-(2k+2)-n, where the n refers to
dimension of input/output dataset and k is the number of dimensions for bottleneck layer. To
achieve robustness, the NLPCA is applied to the tropical zonal wind data (5°S-5°N, 10-hPa to
70-hPa) for a set of k varying from 2 to 5, with 100 runs (different in random initialization
weights) for each k. The optimal number of k is set as 5 as it gives the lowest root-mean-square-
error between the input and output. It is shown that the first and second principal components
(PC1 and PC2) of the NLPCA account for approximately 90% of the whole variance.

Following previous studies (Wallace et al., 1993; Hamilton and Hsieh, 2002; Lu et al., 2009),

the QBO phase index $\psi$ is calculated using PC1 and PC2 as follows:

$$\psi = arctan \, (v/u) \quad (-\pi \leq \psi \leq \pi \, ), \quad (1)$$

where $u$  and $v$ are the time series of the PC1 and PC2, respectively. The positive/negative phase
angle index $\psi$ corresponds to QBOw/QBOe.

We compare the reconstructed zonal wind anomalies using NLPCA and PCA (Wallace et

al., 1993) with the QBO cycle in the observation (Fig. 1). It is shown that the observed QBO
transition corresponds to an abrupt downward propagation in QBOw and a slower downward
transition in QBOe (indicated by clustering points in B to C to A on black triangular shape in
Fig. 1a). The NLPCA captures large part of this sharp transition in QBOw while PCA
underestimates it (indicated by points near C in Fig. 1a). This difference is also clearly shown in
a typical QBO cycle of 1970. 9 – 1972.3 (Figs. 1b, 1b, and 1d, black arrow-sticks exhibits the



downward propagation in QBOw) and the time series of NLPCA/PCA QBO phase (index) (Fig.
S2).
While the NLPCA-derived QBO index is more realistic in following the atmospheric
changes, it is impractical to map the NLPCA phases onto the monthly-mean model diagnostics.
Thus, our QBO composites use simple monthly time steps about our best synchronization point,
which from the NLPCA analysis we take to be the transition from easterlies to westerlies at when
phase angle index $\psi$ crosses 0 with negative values before and positive values after it. It is
demonstrated that compared to QBO composites produced using the PCA-derived QBO index,
that produced using the NLPCA-derived index show larger contrast in observed tropical zonal
wind anomalies between QBOw/QBOe (Supplementary Figs. 3a and 3b) that is consistent with
those described in previous literatures (Hamilton and Hsieh, 2002; Lu et al., 2009). This larger
contrast between NLPCA and PCA in zonal wind anomalies is correspondent with the larger
contrast in that of the total column ozone anomalies (Supplementary Figs. 3c and 3d).

## 4. Linoz calculation of the steady-state ozone

To examine the ozone response to the QBO we use the Linoz model, and the steady-state
ozone is derived from Eq. 4 of Mclinden et al. (2000). The photochemical steady-state ozone
mole fraction $f_{ss}$ (parts per million, moles per mole of dry air) is expressed as follows:
$$f_{ss} = f_o + [\ (P-L)_o + \frac{\partial (P-L)}{\partial T}|_o\ (T-T_o) + \frac{\partial (P-L)}{\partial C_{O_3}}|_o\ (C_{O_3} - C_{O_3}^o)\ ]\tau, \qquad (2)$$

The values $f_o$, $T_o$, and $C_{O_3}^o$ are the climatological values of local ozone, temperature, and
overhead column ozone tables used to calculate the Linoz tendencies. $(P–L)_o$ is the ozone net
production minus loss tendency and the partial derivatives are the sensitivity of the net
production to temperature and overhead column ozone. All of these quantities are evaluated at
the climatological values and tabulated by Linoz as a function of month, pressure altitude, and
latitude. The effective lifetime of ozone, $\tau$, is calculated from the Linoz tables as the negative
reciprocal of the tabulated partial derivative of the production with respect to ozone, i.e., $\tau = $
$-[\frac{\partial (P-L)}{\partial f}|_o\ ]^{-1}$). A major assumption here is that the key chemical families (NOy, Cly, Bry)
and long-lived reactive gases (N$_2$O, CH$_4$, H$_2$O) do not change from their climatological values
used to generate the tables (Hsu and Prather, 2009). This steady-state calculation ignores
transport tendencies and thus will be apply only where the photochemistry is rapid, $\tau < 100$ days.



In application, we derive $f_{SS}$ first locally from the T profile, and then calculate $C_{O_3}$ to correct
for the column ozone sensitivity. Note the calculation of $C_{O_3}$ includes the column ozone based
on the $f_{ss}$ values from all the layers overhead plus a contribution from the local $f_{SS}$ in that layer
weighted by the air mass in the upper half of the layer. Thus, equation 2 becomes a linear
algebraic equation involving $f_{SS}$. Fig. 2 shows this steady-state calculation ($f_{SS}$, T, τ) for January
and July using ERA5 monthly mean temperature.

## 5. Impact of QBO on stratospheric ozone

Nudging the tropical zonal wind creates QBO-driven perturbations to the temperature and
residual circulation that we can diagnose in both the E3SMv2 and CESM2 runs and compare
with observations. For E3SMv2 with interactive ozone we are able to see the changes in ozone.
This also applies to the simulations with an internally generated QBO.
We create a similar composite of the QBO cycle using E3SMv2/CESM2 following Ruiz et
al., (2021) to see the full QBO cycle influence on stratospheric ozone. The time-composite is
created for each month starting 14 months prior and extending to 14 months after the QBO
transition for 1979-2020. The center is when the NLPCA-derived QBO phase angle index (see
section 3) shifts from negative to positive (QBOe -> QBOw). We create the total column ozone
(TCO), tropical ozone, and extratropical ozone composite as a function of QBO phase. For the
TCO, we calculate the zonal-mean averages to produce the global map of composite. For tropical
(15°S-15°N) and extratropical (30°S-60°S/30°N-60°N) ozone, the data is processed to produce
vertical profiles of regional average ozone using latitudinal weight to produce the vertical profile
composite. The CESM2 ozone composite is not shown since its ozone is prescribed. To further
analyze the impact of QBO-induced circulation on ozone, the process is also repeated for
temperature, zonal wind and steady state ozone (see section 4). We first analyze the impact of
QBO on global TCO in section 5.1, and separately analyze impact on tropical and extratropical
stratospheric ozone in section 5.2 and 5.3, followed by the overall performance in section 5.4.

### 5.1 Impact of QBO on global TCO

In this section, we examine the impact of QBO on ozone using TCO observations (MSRv2)
and E3SMv2 model simulations. The TCO composites from E3SMv2 nudged and natural
simulations are compared in Fig. 3.





It is shown that the anomalous MSRv2 TCO exhibits a significant monopole-to-tripole
pattern from QBOe to QBOw (Fig. 3a). The TCO pattern exhibits a monopole pattern of
anomalous low during QBOe that gradually transits to tri-pole pattern of anomalous high in the
tropics and low in the extratropic. The magnitude of the negative in QBOe (5 DU) is lower than
the positive pattern (12 DU) in QBOw in the tropics, indicating asymmetric response of TCO to
QBO in the tropics. The E3SMv2 nudged simulation is like MSRv2 in that it captures most of
the monopole-to-tripole pattern within the tropics and extratropics with similar amplitudes (Fig.
3b), indicating the impact of nudged QBO on TCO is close to what observed. Internally
generated QBO variability in E3SMv2 natural, on the other hand, only partly exhibits the
patterns of MSRv2 (Fig. 3c) with weaker amplitude (nearly eight times weaker). This indicates
the QBO-related signal is partly present in natural E3SMv2, and that nudging the tropical zonal
wind contributes to the modulation and enhancement of this "QBO-driven" TCO variability.

### 283     5.2 Impact of QBO on tropical stratospheric ozone

In this section, we analyze the impact of QBO on tropical (15°S-15°N) stratospheric ozone
concentration. The composites of ozone vertical profile (1-hpa to 100-hPa) from E3SMv2
nudged and natural simulations are compared with the CMZM satellite data (Fig. 4).
It is shown that the CMZM satellite ozone exhibits a double-peak vertical structure with large
ozone variations between 1~20-hPa and 20~100-hPa (Fig. 4a). Both peaks shift in a sequence of
negative-positive-negative from QBOe to QBOw, and the amplitude of the upper peak is smaller
than that of the lower peak (Fig. 4a). The E3SMv2 nudged simulation captures parts of the
double-peak structure (Fig. 4b). The E3SMv2 natural simulations, on the other hand, show
similar double-peaked patterns but with smaller amplitude (3 times weaker) and shorter period
(Fig. 4c). This may be because the period of internally generated QBO in E3SMv2 is ~21 years
(Golaz et al., 2022). Overall, the E3SMv2 nudged simulation modifies the period and enhances
the QBO response in ozone that is mostly consistent with the CMZM weaker above 20-hPa and
stronger below 20-hPa.
As a coupled system, the QBO chemical and transport impacts on ozone are intertwined,
making it difficult to diagnose which QBO impact is more important to the ozone differences
between model and observation or among different models. Here we try to quantitatively
separate these two terms with a new diagnostic tool, recognizing their time scale differences. We
derive the steady state ozone (see Section 4 for details) for E3SM nudged and natural simulations



(Figs. 5a and 5c). The steady state ozone for CESM2 is also derived (Figs. 5b and 5d). Although
ozone is prescribed in CESM2, the steady state ozone for CESM2 shown here is the "would-be"
temperature-ozone if CESM2 were to implement Linoz v2 as its diagnostic ozone module. The
steady state ozone in both E3SMv2 and CESM2 nudged simulations show similar patterns of
ozone peak above 20-hPa while weak response below 20-hPa (Figs. 5a and 5b). The steady state
ozone of E3SMv2 and CESM2 natural simulations (Figs. 5c and 5d) partly resemble that of the
nudged simulations except with weaker amplitude and different periods (shorter for E3SMv2 and
longer for CESM2). This corresponds to their similar alternating temperature pattern phase shift
in the tropics (Figs. 6b, 6c, 6d and 6e) and indicates that the QBO impacts ozone through
temperature-sensitive, fast chemical reactions above 20-hPa. The prognostic ozone in E3SMv2
below 20-hPa corresponds to the alternating $w^*$ shift patterns (Figs. 6g and 6i). This and the no
response in steady state ozone indicates the prognostic ozone below 20-hPa in E3SMv2 is
transport-driven. This demarcation of the QBO-induced ozone at 20-hPa may be due to the
separation of ozone lifetime below/above 20-hPa (Reed et al., 1964). The ozone lifetime is
relatively long compared with the dynamical process below this level while shortened
considerably above it. The temperature affects ozone above 20-hPa through ozone destruction –
colder/warmer anomalies slow/accelerate ozone destruction, leading to correspondent ozone
increase/decrease (Wang et al., 2022); the transport effect of QBO-related wind modulates the
temperature through thermal wind balance enhancing/lessening the upward motion in the tropics
(Plumb and Bell, 1982; Baldwin et al., 2001; Ribera et al., 2004; Punge et al., 2009). This
explains the apparent separation of transport- and chemistry-driven ozone changes above/below
20-hPa. It is also worth mentioning that the nudged CESM2 also produces similar temperature
and $w^*$ (Figs. 6c and 6h), it thus indicates that nudged CESM2 may produce similar prognostic
ozone if it were to implement Linoz v2 as interactive ozone module. Overall, there are apparent
demarcation of QBO impact on tropical stratospheric ozone (15°S-15°N) above/below 20-hPa in
the nudged runs that can separately be explained by transport and chemical impact.
**5.3 Impact of QBO on extratropical stratospheric ozone**
We extend the analysis of the impact of QBO on ozone to the extratropical region in both
hemispheres (30°N-60°N/30°S-60°S). Since in the nudged simulations the nudging is imposed
only in the tropical regions, we can further examine the impact of nudged QBO in the
extratropics where it is free running. Fig. 7 shows pressure-time cross-section of the extratropical



(30°N-60°N/30°S-60°S) ozone concentration as a function of QBO phase for CMZM satellite
ozone, E3SMv2 nudged, and E3SMv2 natural simulations. It is shown that nudged E3SMv2
simulations follow the similar positive-negative ozone phase shift in both hemispheres (Figs. 7b
and 7e). The difference is that ozone is slightly stronger in QBOe while similar amplitude in
QBOw. The natural E3SMv2 simulation does not reproduce the patterns of the nudged
simulation for both hemispheres (Figs. 7c and 7f). This indicates that the nudged QBO is driving
the phase shift of E3SMv2 ozone in both hemispheres' extratropic. For the natural simulations,
the deficiency is likely due to the weak internally generated QBO in E3SMv2. Overall, the
nudged E3SMv2 captures the QBO signal propagated outside of tropics and produces the
extratropical ozone phase shift in both hemispheres. The natural simulation does not show the
phase shift potentially due to weaker internally generated QBO.

In terms of the transport/chemical impact separation, we follow the analysis of Fig. 5 for

E3SMv2 and CESM2 using the Linoz v2 model (Fig. 8). Like that of the analysis in the tropics,
the chemical impact is stronger above 20-hPa for both E3SMv2 and CESM2 nudged simulations
(Figs. 8a, 8b, 8e, and 8f), except the Southern Hemisphere (30°S-60°S) is overall noisier than
that of the northern hemisphere (30°N-60°N). The natural simulations between the two models
are different. The E3SMv2 natural simulations generally show consistent negative phase (Figs.
8c and 8g). The CESM2 natural simulations exhibits similar pattern to the nudged simulations in
the northern hemisphere while that in the Southern Hemisphere is noisier (Figs. 8d and 8h). This
noisier southern hemisphere steady state ozone above 20-hPa in the nudged simulations
correspond to the noisier temperature for the two models (Figs. 9g and 9h), which may be largely
affected by stronger and noisier southern polar vortex (Supplementary Figs. 5a and 5b) as also
documented by other studies (Ribera et al., 2004). The intrusion of the polar vortex via events
like stratospheric sudden warming (Butler et al., 2017) may have an impact on the QBO-ozone
relationship in the extratropics. Below 20-hPa, the E3SMv2 nudged ozone corresponds to the $\underline{w}^*$
(Fig. 9l and 9q), indicating it's transport-driven.

In terms of the impact of QBO nudging in the extratropics, there are considerable differences

between the two models especially in the Southern Hemisphere. It shown that CESM2's phase
shift of temperature and $\underline{w}^*$ patterns (Figs. 9h and 9r) in the Southern Hemisphere is not as
obvious as that shown in E3SM (Figs. 9g and 9q). Since this is outside the nudging region, it's
complicated to differentiate the main impact factor. One reason may be the different ozone

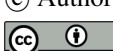



feedback between the two models – interactive ozone in E3SMv2 contributes to maintain the
QBO-temperature structure (Butchart et al., 2023) while prescribed ozone in CESM2 does not;
Another may be due to the overall 3-D nudging strategy in E3SMv2 that may provide more
stringent constraint than the 2-D zonal mean nudging strategy that CESM2 adopted. Another
interesting issue is that both nudged simulations can partly reproduce the observed ERA5
temperature and $w^*$ patterns in the extratropic regions outside of the tropical nudging regions.
This occurrence of the residual circulation consistent with the ERA5 in the extratropics indicates
the validity of the nudging strategy for the QBOi protocol. Despite the fixed ozone, the CESM2
could still produce such circulation and temperature patterns that is consistent with ERA5
indicates the overall weak ozone feedback on formation of circulations both in the tropics and
extratropics. These patterns and the steady state ozone analysis for the CESM2 nudged
simulation also indicate that it may reproduce the prognostic ozone like E3SMv2 if it were to use
Linoz-v2 as interactive ozone module under the QBOi nudging protocol.
**5.4 Model performance in simulating QBO impact**

In this sub-section, we examine the overall performance of E3SMv2 and CESM2 QBOi

simulations in simulating the QBO-ozone relationship. We evaluate the pattern correlation and
standard deviation of the area-weighted TCO pattern (60°S-60°N), vertically-weighted ozone
concentration (15°S-15°N, 30°N-60°N, 30°S-60°S), zonal wind (15°S-15°N, 30°N-60°N, 30°S-
60°S), temperature (15°S-15°N, 30°N-60°N, 30°S-60°S), and $w^*$ (15°S-15°N, 30°N-60°N, 30°S-
60°S). For ozone, only E3SMv2 results are shown since CESM2 has fixed ozone. The results are
summarized in a Taylor diagram shown in Fig. 10. The observed pattern is plotted at the (1,0)
reference point.

In terms of ozone (Fig. 10a), there are remarkable differences between the simulations.

Overall, the E3SMSv2 nudged simulations perform the best, with the pattern correlation of all
four variables over 0.8 while other simulations are below 0.5. This indicates nudging realistic
QBO variability may increase the model performance in simulating ozone. In the extratropics,
the E3SMv2 nudged simulation has good pattern correlations, but the amplitude is off by over
1.5 times. The results for temperature, zonal wind and $w^*$ are similar with ozone in the tropics
(Figs. 10b, 10c, and 10d). What's different is in the extratropics — both nudged
E3SMv2/CESM2 temperature, zonal wind, and $w^*$ show better performance in NH extratropics



than in SH extratropics. This may be due to stronger polar vortices in SH and NH that disturb the
QBO signal. Another difference is in the natural simulations – the tropical temperature (15°S-
15°N) and zonal wind signals exhibit reasonable correlations of over 0.7 in zonal wind, over 0.5
in temperature. This indicates a discernable internally generated QBO signal in the
E3SMv2/CESM2, although it's weaker and does not extend to the extratropics.

## 6. Discussion and Conclusion

### 6.1 Discussion

There are some interesting issues worth mentioning in this study. The first is the effect of

nudging. It is shown that even in the extratropical regions where the QBO nudging is not
imposed, the QBO impact on extratropical circulation is still apparent in the two models. In these
QBOi simulations, the E3SMv2 employs a 3-D nudging strategy where the ERA5 3-D full field
zonal wind is nudged to the model while CESM2 employs a 2-D nudging strategy. It may be
recognized that the E3SMv2 posed a stronger nudging than CESM2, but both strategies were
able to produce extratropical QBO-associated circulation outside of the nudging region, this
demonstrates the overall effectiveness of the nudging strategy. Between the models, there are
still minor differences. For example, the extratropical zonal wind and temperature in CESM2 are
more scattered than that of E3SMv2. One reason may be the nudging strategy discussed above,
another reason may be the interactive/non-interactive ozone in the model. In this QBOi
simulation setup, E3SMv2 has the interactive ozone turned on, while CESM has only fixed
ozone input. Thus, the QBO-ozone interaction in E3SMv2 may be more self-consistent than that
in CESM2 – Studies have documented the impact of QBO-ozone interaction tend to maintain
and the QBO-temperature structure and prolong its period (Hasebe et al., 1994; Shibata, 2021;
Butchart et al., 2023).

Another noteworthy issue is the use of the offline Linoz v2 model to diagnose the dynamic

and chemical impact of QBO on ozone. It is demonstrated here that the Linoz v2 is a simple but
useful tool to diagnose and separate the dynamic/chemical impact of QBO on ozone. The results
shown here are important for diagnosing model-model and model-observation differences in the
QBO-ozone responses for climate simulations.

### 6.2 Conclusion



In this study, we utilize the Linoz v2 model to separate the chemical and transport response
of the QBO ozone impact in climate models. We derive a new QBO phase index using an
NLPCA method, and utilize the index to form QBO cycle composites to analyze QBO-ozone
relationship in observation and simulations produced under the QBOi protocol. By analyzing the
simulations of two QBOi participant models (E3SMv2 and CESM2), it is shown that the nudged
E3SMv2 simulation captures the monopole-to-tripole composite pattern in the observed TCO.
The natural simulation partly reproduces the observed TCO pattern but with weaker amplitude
and shorter period, indicating there is an internally generated QBO in E3SMv2 that is enhanced
and prolonged by nudging in the E3SMv2 nudged simulations. Looking further into the vertical
structure of the QBO-ozone relationship, it is shown that the E3SMv2 nudged simulations
capture most of the double-peaked vertical structure in observed ozone data between 1~20-hPa
and 20~100-hPa in the tropics but with weaker amplitudes in the extratropics. natural simulation
only captures part of the structure with smaller amplitude, indicating the existence of internally
generated QBO. This and the nudged simulations indicate that nudging enhanced the QBO
amplitude and prolonged its period originally exists within E3SMv2.
Utilizing the Linoz v2, we separated the chemical and transport response of ozone in
E3SMv2 nudged to QBO. It is shown that the two impacts have a rather clear demarcation on
both tropical and extratropical ozone response below/above 20-hPa – chemistry impact
correspondent to QBO-related temperature change dominates the response above 20-hPa linked
to photochemical process, and transport impact related to QBO-related vertical motion dominates
the response below 20-hPa. The results here are important for diagnosing model-model and
model-observation differences in the QBO with free-running climate-change simulations,
allowing us to separate temperature from circulation effects. In CESM2, the fixed ozone that is
out-of-phase with the observed QBO variability seems to impose a weak constraint on the overall
simulation. This indicates that using interactive ozone or not in the simulation does not
significantly alter the results for QBO simulations, although the synchronization impact of QBO
variability in observed ozone may need further examination (Butchart et al., 2023).
Stratospheric ozone is not only essential for protecting life on the Earth but also has
important climate impacts. More and more studies reported the important role of ozone
variations in modifying the stratospheric circulation and therefore influencing the surface climate
(e.g. Xie et al., 2020). Since the QBO has relatively high predictability, considering its impacts



on stratospheric ozone and subsequent atmospheric circulations may help improve the
predictions of surface weather and climate (e.g., Li et al., 2023).
Despite the above studies, however, there are still caveats. Firstly, the current study makes
use of only one model in QBOi that has interactive ozone feature. More models may be used in
the future to examine the QBO-ozone relationship. The capability of the current version of O3v2
in E3SMv2 is limited due to the missing representation of the $NO_x$ long-lived tracers. In the
latest version of E3SMv3, the O3v2 is updated to include the impacts of these tracers, it would
be interesting to see how these tracers interact with the current ozone calculations. These are to
be assessed in future studies.

## 464 Data availability

The satellite data from the Copernicus Climate Change Service can be accessed at
(https://cds.climate.copernicus.eu/cdsapp#!/dataset/satellite-ozone-v1?tab=form). The ERA5
data can be accessed at (https://cds.climate.copernicus.eu/cdsapp#!/dataset/reanalysis-era5-
complete?tab=overview). The ChemDyg diagnostics can be accessed at
(https://doi.org/10.5281/zenodo.11166488).


## 472 Author contribution

J.X., Q.T. designed the research; J.X. performed the E3SM simulations and wrote the
manuscript. J.R. provided the CESM2 simulation. Q. T. and M.P.'s supervised the research and
helped interpreting the results. All authors contributed to the scientific discussion and paper
revision.

## 478 Competing interests

The authors declare that they have no conflict of interest.


## 482 Acknowledgement



We thank the Copernicus Climate Change Service for providing the satellite data  and ECMWF
for the ERA5 data. We thank Isla Simpson for setting up the CESM2 QBOi simulations, and
providing the Python script for generating the Transformed Eulerian Mean variables. We thank
Sasha Glenville for transferring the CESM2 data. This research was supported as part of the
E3SM project, funded by the U.S. Department of Energy, Office of Science, Office of Biological
and Environmental Research. Part of the work was supported by the LLNL LDRD project 22-
ERD-008 titled "Multiscale Wildfire Simulation Framework and Remote Sensing". E3SM
simulations were performed on a high-performance computing cluster provided by the BER
ESM program and operated by the Laboratory Computing Resource Center at Argonne National
Laboratory. Additional post-processing and data archiving of production simulations used
resources of the National Energy Research Scientific Computing Center (NERSC), a DOE
Office of Science User Facility supported by the Office of Science of the U.S. Department of
Energy under Contract No. DE-AC02-05CH11231. This work was performed under the auspices
of the U.S. Department of Energy by Lawrence Livermore National Laboratory under contract
DE-AC52-07NA27344. The IM release number is LLNL-JRNL-858987. This work was in part
supported by the National Center for Atmospheric Research (NCAR), which is a major facility
sponsored by the National Science Foundation (NSF) under Cooperative Agreement 1852977.
Portions of this study were supported by the Regional and Global Model Analysis (RGMA)
component of the Earth and Environmental System Modeling Program of the U.S. Department of
Energy's Office of Biological and Environmental Research (BER) via NSF Interagency
Agreement 1844590.



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





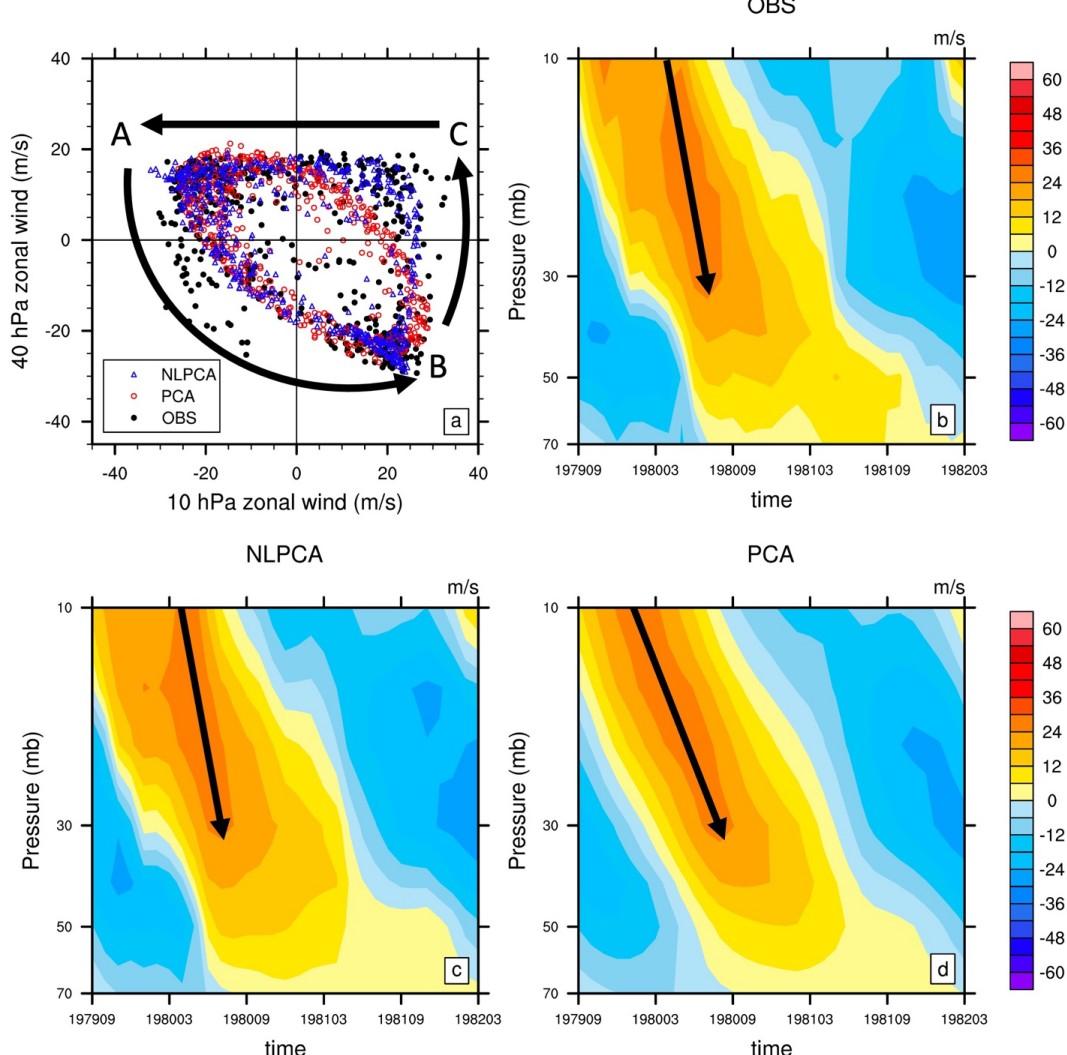


Fig. 1 (a) Scatter-plot of 1979-2020 anomalous monthly mean zonal wind (m/s) at 10-hPa vs 40-
hPa for observation (black), NLPCA reconstruction (blue), and PCA reconstruction (red).
Typical cycle of QBO from (b) Observational station data from University of Berlin, (c) NLPCA
reconstruction, (d) PCA reconstruction.







Fig. 2 The (a, b) steady state ozone (mol/mol) derived using LINOZ on E3SMv2 temperature, (c,
d) ERA5 temperature (°C), (e, f) photochemical relaxation time $\tau$ (days), and for January and
July. The thick black line in (c, d) denotes the 300 value-line.



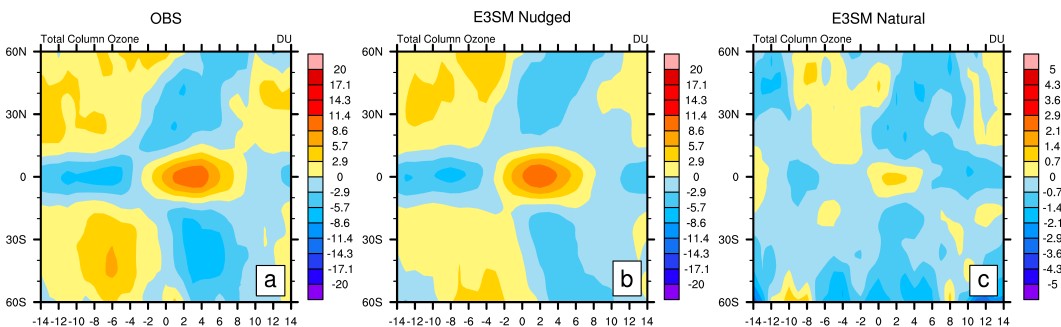


Fig. 3 Total column ozone (TCO, Dobson Unit) anomaly (relative to 1979-2020 mean)
composites as function of QBO phase (determined by NLPCA QBO index) for (a) OBS (Multi-
Sensor Reanalysis version 2), (b) E3SMv2 nudged simulation, (c) E3SMv2 natural simulation. 0
is centered on the month when QBO transits from QBOe to QBOw (determined by when current
QBO index<0 and next QBO index>0). The QBO phase is determined by 5S-5N average of the
zonal wind.




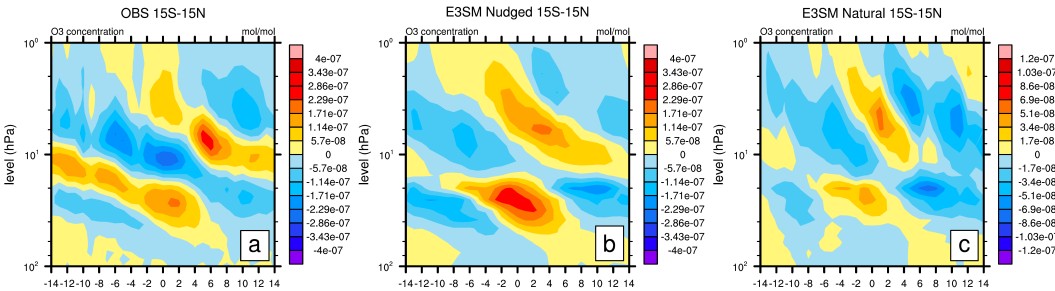


Fig. 4 Pressure-time cross-section of the tropical (15°S-15°N) 1979-2020 anomalous ozone
concentration (mol/mol) as function of QBO phase for (a) OBS (Concentration Monthly Zonal
Mean), (b) E3SMv2 nudged simulation, (c) E3SMv2 natural simulation. 0 is centered on the
month when QBO transits from QBOe to QBOw (determined by when current QBO index<0
and next QBO index>0). The QBO phase is determined by 5S-5N average of the zonal wind.


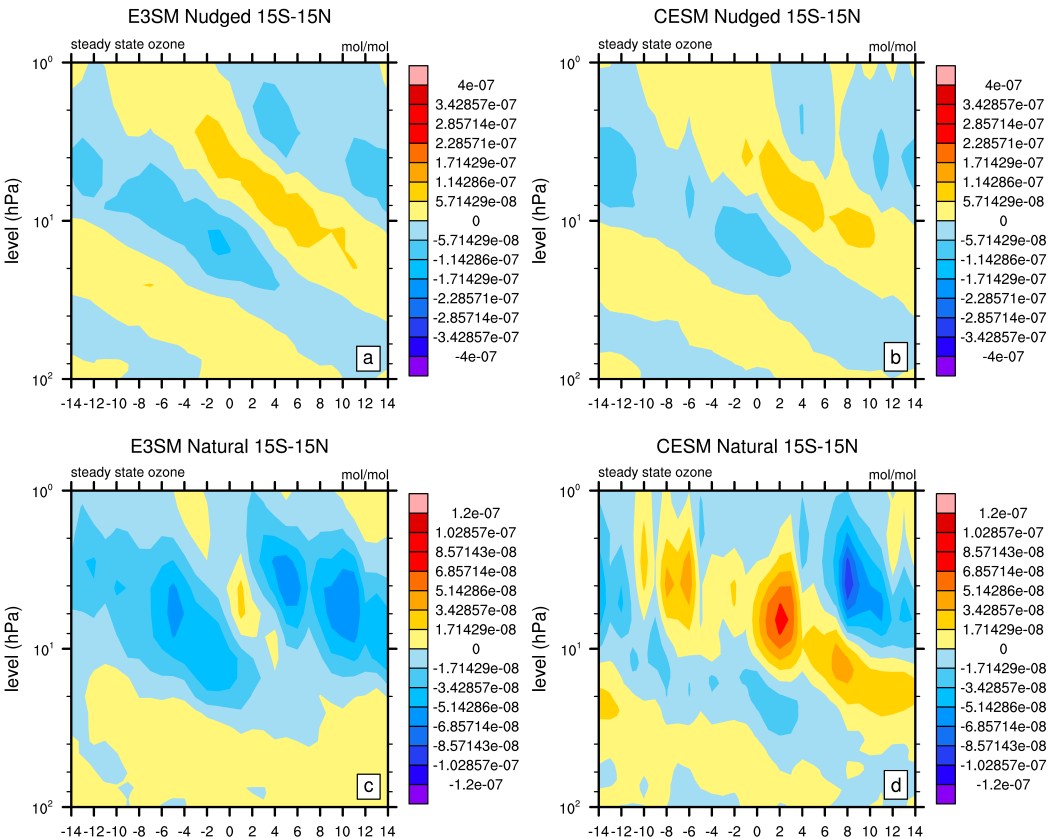

Fig. 5 Pressure-time cross-section of the tropical (15°S-15°N) 1979-2020 anomalous Linoz
steady state ozone (mol/mol) as function of QBO phase for (a, c) E3SMv2 and (c, d) CESM2. 0
is centered on the month when QBO transits from QBOe to QBOw (determined by when current
QBO index<0 and next QBO index>0). The QBO phase is determined by 5S-5N average of the
zonal wind.




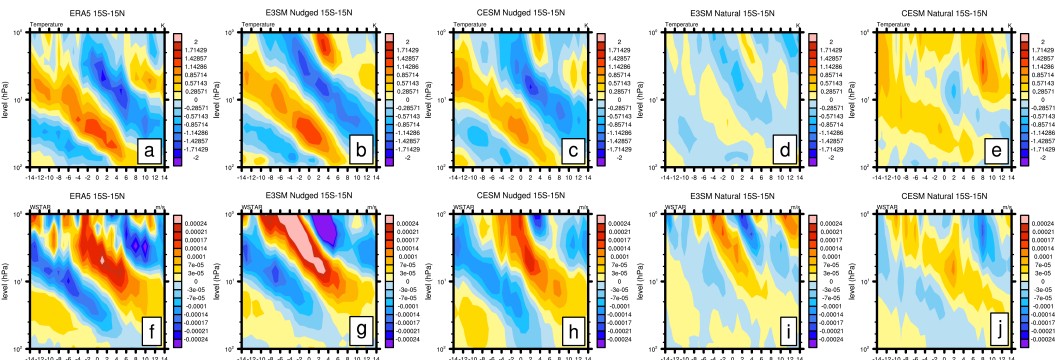

Fig. 6 Pressure-time cross-section of the tropical (15°S-15°N) 1979-2020 anomalous temperature (K) and $\underline{w}^*$ (Transformed Eulerian Mean residual vertical transport, m/s) as function of QBO phase for (a, f) ERA5, (b, d, g, i) E3SMv2, (c, e, QBOe, j) CESM2. 0 is centered on the month when QBO transits from QBOe to QBOw (determined by when current QBO index<0 and next QBO index>0). The QBO phase is determined by 5S-5N average of the zonal wind.

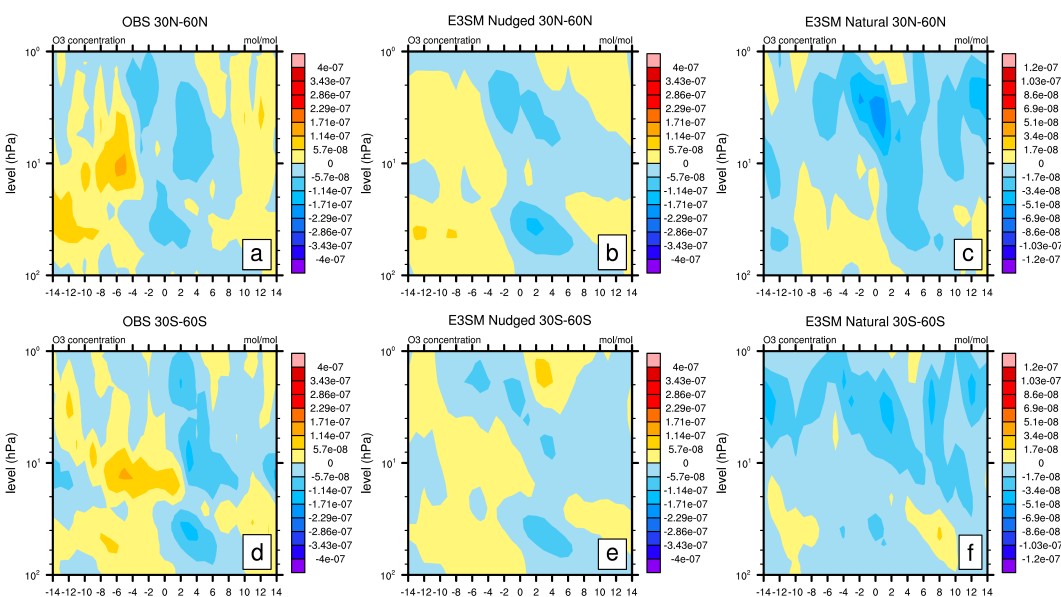

Fig. 7 Pressure-time cross-section of the extratropical (30°N-60°N/30°S-60°S) 1979-2020

anomalous ozone concentration (mol/mol) as function of QBO phase for (a, d) OBS (CMZM),

(b, e) E3SMv2, (c, f) CESM2. 0 is centered on the month when QBO transits from QBOe to

QBOw (determined by when current QBO index<0 and next QBO index>0). The QBO phase is

determined by 5S-5N average of the zonal wind.

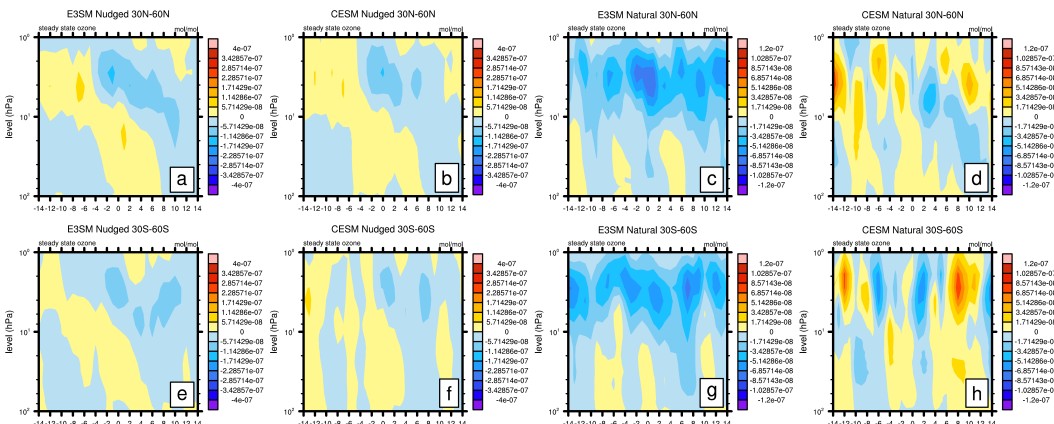

Fig. 8 Pressure-time cross-section of the extratropical (30°N-60°N/30°S-60°S) 1979-2020 anomalous steady state ozone (mol/mol) as function of QBO for (a, e) E3SMv2 nudged, (b, f) CESM2 nudged, (c, g) E3SMv2 natural, (d, QBOe) CESM2 natural. 0 is centered on the month when QBO transits from QBOe to QBOw (determined by when current QBO index<0 and next QBO index>0). The QBO phase is determined by 5S-5N average of the zonal wind.

881

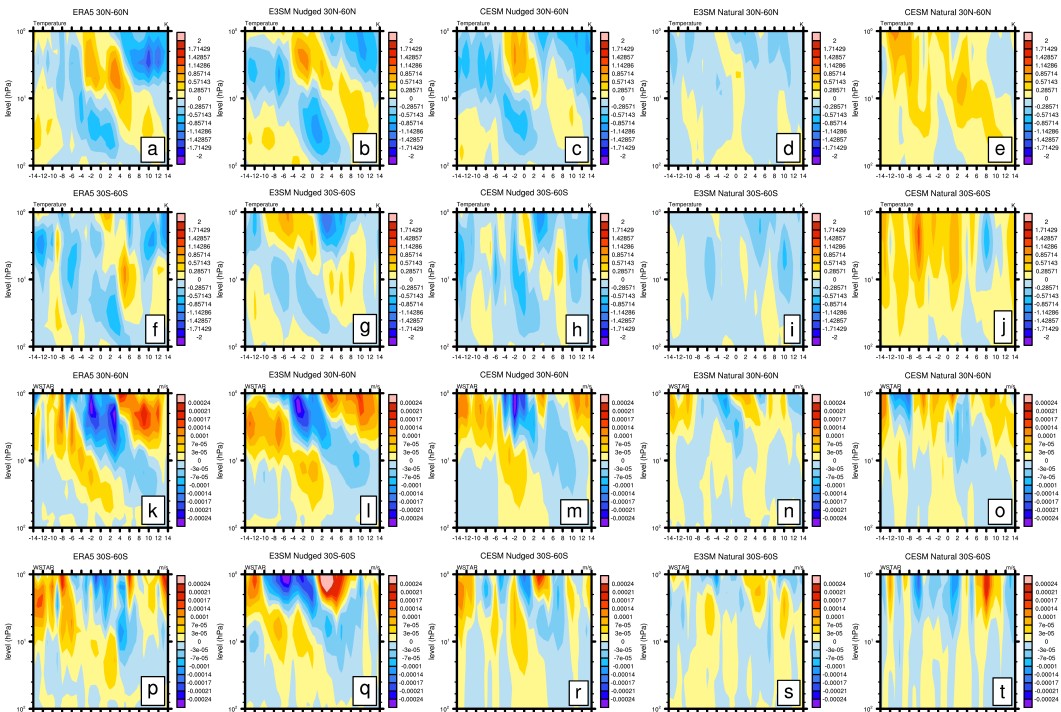

882

Fig. 9 Pressure-time cross-section of the extratropical (30°N-60°N/30°S-60°S) 1979-2020

temperature (K)/ $\underline{w}^*$ (m/s) as function of QBO for (a, f, k, p) ERA5, (b, g, l, q) E3SMv2 nudged,

(c, QBOe , m, r) CESM2 nudged, (d, i, n, s) E3SMv2 natural, (e, j, o, t) CESM2 natural. CESM2.

0 is centered on the month when QBO transits from QBOe to QBOw (determined by when

current QBO index<0 and next QBO index>0). The QBO phase is determined by 5S-5N average

of the zonal wind.

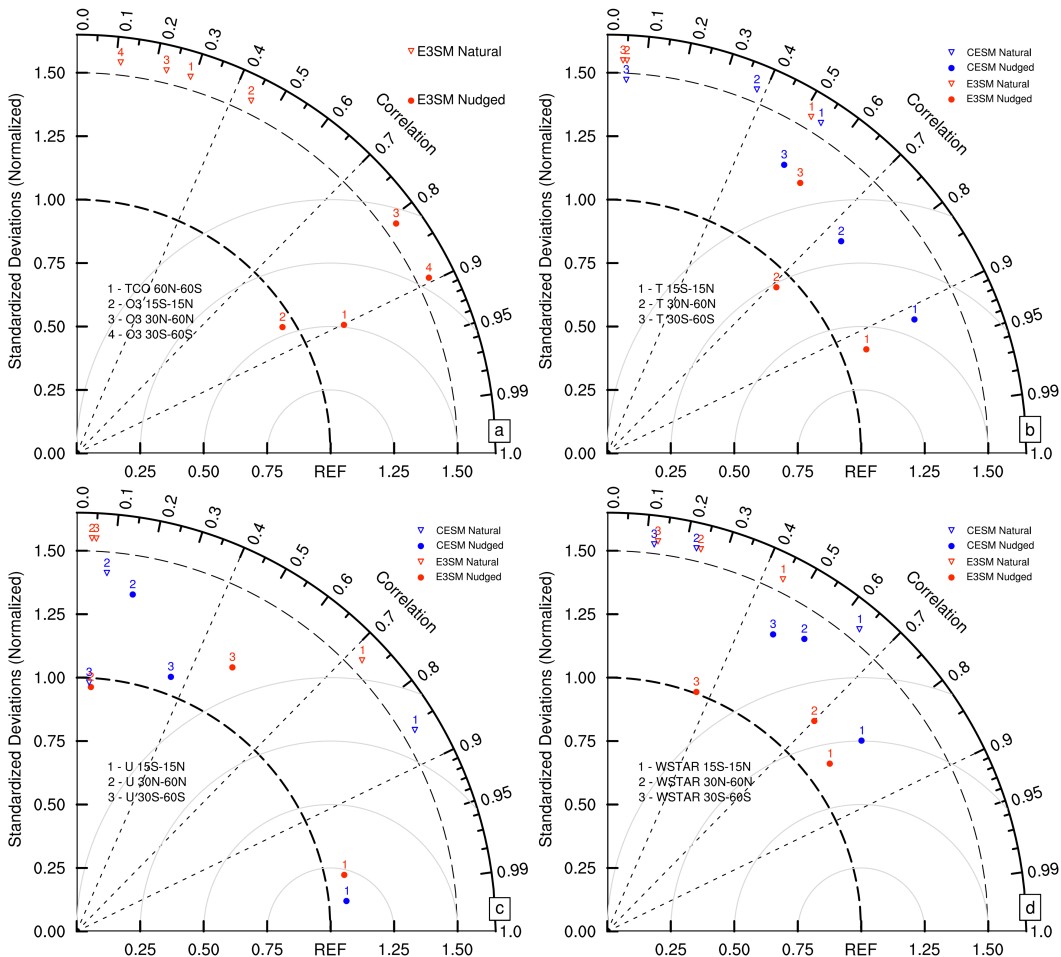

Fig. 10 Taylor diagram of the E3SMv2/CESM2 simulation for various datasets for 1979-2020.

(a) The area-weighted total column ozone (60°S-60°N, DU) and pressure-time cross-sections of

ozone concentration (15°S-15°N, 30°N-60°N, 30°S-60°S, mol/mol) anomalies with OBS (MSR

and CMZM), respectively. (b) The area-weighted pressure-time cross-sections of temperature

(15°S-15°N, 30°N-60°N, 30°S-60°S, K) anomalies with ERA5. (c) The area-weighted pressure-

time cross-sections of zonal wind (15°S-15°N, 30°N-60°N, 30°S-60°S, m/s) anomalies with

ERA5. For pattern correlations, the cross-sections are weighted by pressure layer thickness. On

all Taylor diagrams, the model standard deviations are normalized by dividing the standard

deviations of the reference.





