# Peer review of "Disentangling the chemistry and transport impacts of the Quasi-Biennial Oscillation on stratospheric ozone"

_EGUsphere, 2024_

## Referee Report (RR1)

Second review of "Disentangling the chemistry and transport impacts of the Quasi-Biennial Oscillation on stratospheric ozone" by Xie et al.

The authors have provided extensive responses to the reviewer comments and have revised their manuscript accordingly. The revision is easier to follow.

Much of the description of the analysis is comprehensive and carefully presented. My recommendation is that this work may be eventually publishable if the authors revise the manuscript to be very clear about what is new from their work and what message the community should learn from this investigation. In particular, the title, abstract, and introduction would benefit from major revision to focus attention on what new can be learned from the model simulations and analysis. As can be seen from the major comments below, I was still confused about the main thrust of the investigation and about how the linear model is used.

Major comments

1. The manuscript still does not do an adequate job of setting up the goals of the project so the reader is left wondering what is the point of evaluating the linear ozone model. Is the eventual goal to use this linear model in interactive runs or is it primarily for purpose of diagnosing ozone? The paper shows that the model with a nudged QBO and expanded range of chemical families (version 3) reproduces ozone reasonably well. The impact of ozone changes due to the QBO on stratospheric or climate variables is not presented and so the manuscript does not make a case for including Linoz as a component of an interactive model. The analysis seems mainly to demonstrate that this simplified model can reproduce results from more comprehensive studies showing that transport of ozone, transport of NOy, and temperature dependence of ozone photochemistry are the dominant processes in the response to the QBO in different vertical regions. The manuscript does not describe any new insights into the chemistry or the impact of the ozone changes on the QBO, rather, it demonstrates that their linear model is able to capture these

2. Unfortunately, even after several times reading through, I'm not sure I was able to follow some basic aspects of the model description. This makes it very difficult to evaluate the results. At line 65-66, "In this study, we use the interactive stratospheric chemistry module in E3SM (Linoz: 65 Mclinden et al., 2000; Hsu and Prather, 2009) as an off-line model …" indicates the Linoz is used offline. But then at line 77-78 "Our primary modeling tool is the Department of Energy (DOE) Energy Exascale Earth 77 Model version 2 (E3SMv2, Golaz et al., 2022) with interactive stratospheric ozone …". And at line 99-100 "Stratospheric ozone in E3SMv2 is calculated interactively through transport and the chemical Linoz module". Then at line 467-468, "the use of the offline Linoz model…" Perhaps part of the problem is the use of the term "interactive" since any potential ozone impact on the QBO is weakened or cancelled by the dynamical nudging.

Other comments

1. Please make it clear what you mean by "phase asymmetry" (e.g. line 74, line 179, etc.). Based on further reading in the manuscript and familiarity with the observations, it appears you mean the difference in the length of time and evolution of the wind change with pressure between the easterly and westerly phases. This should be defined at the outset so the reader is not wondering what asymmetry you are referring to.

2. (l. 326-327) Sentence beginning "This indicates …". Since the response of column ozone is mainly driven by the winds in the lower stratosphere, this discrepancy likely indicates that the internally generated QBO is too weak there.

3. (l. 349-356) What are you trying to say here? The figure indicates poor response of ozone in the simulation with internally generated QBO. What is the "improved representation" that is referred to?

4. Is anything additional learned from the convoluted calculation of the ozone that CESM2 would have if it used Linoz?

Editorial comments

1. (l. 394) "the no response" -> "the lack of response"

2. (l. 427) What does "colder/warmer" mean? It's either one or the other.

3. (l. 469) "with the with the"

---

## Editor Decision (ED1)

**Editor comments on manuscript egusphere-2024-1927 (revised version)**

*Jinbo Xie et al: Disentangling the chemistry and transport impacts of the Quasi-Biennial Oscillation on stratospheric ozone*

**Specific comments:**

P1, L22: It is not clear what is meant with "modeled changes temperature". Please rephrase.

P1, L25: For me it is not clear if these are detected by these changes or if these are caused by these changes.

P2, L35: Add "tropical" before "stratosphere" since this relates to the tropical stratosphere not the entire stratosphere.

P2, L38: Please rephrase the sentence. The QBO itself does not influence the chemical processes in the atmosphere rather the changed composition of the atmosphere due to the QBO is affecting chemistry.

P2, L60: "Nudging of the winds is unphysical and produces an anomalous BDC….". Is this really true. I had a quick look at the Orbe et al. paper and to my understanding they jus stat that care should be taken then SD simulations are compared.

P3, L65ff:  To my knowledge Linoz is providing only very basic, simplified chemistry which I guess is sufficient for climatic studies.  Nevertheless, I would appreciate a statement about the general quality/accuracy of the Linoz scheme.

P3, L83 and 84 and throughout the manuscript: As already stated in my comments before publication in the discussion you should use the Copernicus style and abbreviate write Section beginning with a capital letter  and abbreviate it as  "Sect." unless it appears at the begin of the sentence.

P4, L121: Were tuned ? Please specify this. What exactly has been done?

P5, L135: A reanalysis is no observation even if observations have been assimilated!

P7, L186: "QBOw": I guess you mean here the QBO west. This abbreviation should be introduced.

P8, L220: Please use here a clear writing for dates. Do you mean September 1970 to March 1972?

P9, L255: Please use the Copernicus style. Equation should be abbreviated as Eq. in the text.

P9, L256: Has the abbreviation SSO been introduced?

P9, L266: "The center" not clear. Do you mean the center of the composite is where the NPLA shift is? Please rephrase so that it becomes clear where the center is.

P9, L274-275: Rephrase/Improve sentence.

P10, L287-291: Repetition of what has been said before???

P11, L318: "The TCO patterns exhibits tri-pole pattern of anomalous low in the tropics and high in the extratropics during …." Sentence not clear since something is missing here. Please correct.

P11, L320: "The magnitude of the negative in QBOe"? Not clear. Please correct/rephrase.

P11, L335: Has the abbreviation "CMZM" been introduced?

P12, L341: "low" -> do you mean "low ozone"? Please clearly write this.

P12, L344: Change sentence as follows: Since the nudged E3SMv2 simulation uses the Linoz-v2 scheme where the chemical species such as ….

P12, L347: Change to "with chemistry of NO2-N2O-CH4-H2O included".

P12, L348: Missing parts of what?

P12, L366: Correct as follows: " ……phase changes in the northern hemisphere."

P14, L398: What is noisier? The temperature? Please clearly state this.

P14, L403: Either singular or plural, thus either "like stratospheric sudden warmings" or "like a stratospheric sudden warming". Further "stratospheric" and "sudden" need to be swapped. It should read "sudden stratospheric warming".

P14, L418: "To further examine the variable responsible for the change, the single specie sensitivity….." Please improve sentence.

P14, L426: Please clearly state which level.

P15, L431-432: Check sentence and rephrase. Add "that is" before controlled? What is meant with "return arm".

P16, L441: Similar than what? Please use clear statements.

P15, L442: Change to "…. Similar prognostic ozone if Linoz were to be implemented….."

P14 and P15: References to the figures are given in a very confusing order. You are jumping here back and forth between the figures which makes it very hard to follow.

P15, L455: Here references should be given. I assume this statement is based on results from previous studies.

P16, L459ff: NH -> northern hemisphere, SH -> southern hemisphere. Please use a consistent writing.

P16, L460: Also here references should be added.

P16, L487: "less field of high wavenumbers" not clear

P17, 489: "with nudging" -> please rephrase

P17, L502: Check sentence and please correct.

Supplement: Check numbering of figures. Figure S7 appears twice.

**Figure captions in general:**

These should start with "Figure" and the respective number followed by a colon. "Figure" is only abbreviated in the text!

anomalous -> anomaly

use a readable date formats.

Add degree sign to the given coordinates.

Figure 8 caption: OBS? Do you mean "observations"?

**Technical corrections:**

P2, L33: add "atmospheric" before "circulation".

P2, L33: that -> and

P2, L55: Write "zero" instead of "0".

P4, L104: about -> of (?)

P6, L179: QBOs -> QBO

P10, L287: tropics -> tropical

P10, L302: add "the" -> the northern

P10, L303: accordance? Do you mean "agreement"?

P10, L305: extratropics -> extratropical

P11, L307: is -> are

P11, L310: creates more -> creates a more

P11, L334: extratropics -> extratropical

P12, L345: chemistry -> chemical

P12, L354: add "being" so that it reads "being weaker".

P12, L355: add "the" -> the E3SM-v2

P13, L367:  hemisphere -> hemispheres

P13, L375: section 4 -> Sect. 4

P13, L374: equation 2 and 3 -> Eq. 2 and 3

P13, L389 and 392: correspond -> corresponds.

P14, L398: Southern Hemisphere -> southern hemisphere

P15, L439: process -> processes

P16, L483: Add "the" so that it reads "by the two models".

P16, L439: change -> changes

P16, L462: write "and" instead of using a comma.

P16, L465: Conclusion -> conclusion

P16, L467: Firstly -> First

P16, L468: dynamic -> dynamical

P16, L468: add "the" so that it reads "of the QBO".

P16, L469: delete "while the".

P16, L470: profile -> profiles (?)

P16, L 471: Delete "the"

P16, L475: difference -> different

P16, L482: degree sign missing.

P17, L488: Lastly -> Last

P17, L507 and 509: Add "the" so that it reads "the nudged" and "the QBO", respectively.

P17, L512: process -> processes

P17, L515: Delete "change".

---

## Author Response (AR2)

Item-by-item response to all review comments

NOTE: To facilitate the evaluation of our responses, original review comments are listed first in their originals (in black), followed by our itemized response (in blue). An annotated version of revised manuscript is attached.

We thank the reviewers' comments, which are in black text below. Our response is followed (in blue).
* * *
We thank the reviewers' comments on helping to improve this manuscript. We have made substantial modifications about the new contributions of the current study to address the reviewer's concerns as follows:

1. Novel diagnostic tool
   The current paper develops a novel offline diagnostic tool of steady-state ozone (SSO) to be applied to climate models to separate the model-to-model discrepancy in QBO-ozone due to chemical and transport process.
   Please refer to section 1, section 5.4 and 6.1 for further detail.

2. Improved ozone representation at negligible cost
   Nudged QBOi experiments with improved linearized ozone chemistry ($NO_y$-$N_2O$-$CH_4$-$H_2O$) may improve QBO-ozone simulation than that without. This demonstrates that online linearized ozone (Linoz) calculation with negligible computational cost can be a useful tool for studying ozone and QBO-ozone relationship.
   Please refer to section 5.3 and 6.1 for further detail.

3. Ozone damping effect on QBO
   Additional nudged E3SMv2 simulations indicate that fixed ozone shown stronger QBO-temperature, indicating damping effect of ozone on QBO.
   Please refer to last paragraph of section 6.1 for further detail.
* * *
Reviewer 1

The authors have provided extensive responses to the reviewer comments and have revised their manuscript accordingly. The revision is easier to follow.
Much of the description of the analysis is comprehensive and carefully presented. My recommendation is that this work may be eventually publishable if the authors revise the manuscript to be very clear about what is new from their work and what message the community should learn from this investigation. In particular, the title, abstract, and introduction would benefit from major revision to focus attention on what new can be learned from the model simulations and analysis. As can be seen from the major comments below, I was still confused about the main thrust of the investigation and about how the linear model is used.

Major comments
1. The manuscript still does not do an adequate job of setting up the goals of the project so the reader is left wondering what is the point of evaluating the linear ozone model. Is the

eventual goal to use this linear model in interactive runs or is it primarily for purpose of diagnosing ozone? The paper shows that the model with a nudged QBO and expanded range of chemical families (version 3) reproduces ozone reasonably well. The impact of ozone changes due to the QBO on stratospheric or climate variables is not presented and so the manuscript does not make a case for including Linoz as a component of an interactive model. The analysis seems mainly to demonstrate that this simplified model can reproduce results from more comprehensive studies showing that transport of ozone, transport of NOy, and temperature dependence of ozone photochemistry are the dominant processes in the response to the QBO in different vertical regions. The manuscript does not describe any new insights into the chemistry or the impact of the ozone changes on the QBO, rather, it demonstrates that their linear model is able to capture these.

Response: The goal of this paper is to demonstrate the novel use of a steady-state ozone (SSO) metric to separate the chemical and transport impact of QBO-ozone. It is applied to nudged E3SMv2 with interactive ozone chemistry to demonstrate its applicability and further separate the QBO-ozone process with a more realistic QBO simulation.

Nudged E3SMv2 simulations with Linoz-v2/Linoz-v3 exhibit the impact of ozone change due to QBO. These set the base of the QBO-ozone impact, which is applied to SSO for impact separation. We were able to clearly separate the role of temperature, $NO_y$, and transport with pressure, which was still under debate in previous studies. This is added in the text, please refer to section 1, section 5.4, and section 6.1 for further detail.

There are two other new findings are of interest and further discussion added in section 6.1:
1. For one, the inclusion of $NO_y$-$N_2O$-$CH_4$-$H_2O$ chemistry in online Linoz calculation in E3SMv2 shows better QBO-ozone magnitude above 10-hPa than that without. Meraner et al., (2020) demonstrated that the usefulness of linearized ozone in representing ozone due to negligible computational cost, despite its deficiency in simulating QBO-ozone magnitude. The inclusion of $NO_y$-$N_2O$-$CH_4$-$H_2O$ chemistry may contribute to alleviating this issue.
2. For another, the interactive ozone (Linoz calculated online in E3SM and feedback to climate) tends to damp the QBO in E3SMv2. This is indicated by comparing QBO-temperature (Figs. 4, 5 and S8) for E3SMv2 Linoz-v2 simulation with E3SMv2 fixed ozone simulation.
    Please refer to third and fourth paragraph of section 6.1 for further detail.

Overall, the SSO may be used applied to climate model simulations and chemistry climate models (CCM) to provide further insight on model-to-model deficiency. This includes the current QBO initiative (QBOi) simulations, the joint QBOi-CCMI project proposed by The QUasi-Biennial oscillation and Ozone Chemistry interactions in the Atmosphere (QUOCA). We've added this in the text. Please refer to section 6.1 for further detail.

Reference
Meraner, K., Rast, S., & Schmidt, H. (2020). How useful is a linear ozone parameterization for global climate modeling? Journal of Advances in Modeling Earth Systems, 12, e2019MS002003. https://doi.org/10.1029/2019MS002003

2. Unfortunately, even after several times reading through, I'm not sure I was able to follow some basic aspects of the model description. This makes it very difficult to evaluate the results. At line 65-66, "In this study, we use the interactive stratospheric chemistry module in E3SM (Linoz: 65 Mclinden et al., 2000; Hsu and Prather, 2009) as an off-line model …" indicates the Linoz is used offline. But then at line 77-78 "Our primary modeling tool is the Department of Energy (DOE) Energy Exascale Earth 77 Model version 2 (E3SMv2, Golaz et al., 2022) with interactive stratospheric ozone …". And at line 99-100 "Stratospheric ozone in E3SMv2 is calculated interactively through transport and the chemical Linoz module". Then at line 467-468, "the use of the offline Linoz model…" Perhaps part of the problem is the use of the term "interactive" since any potential ozone impact on the QBO is weakened or cancelled by the dynamical nudging.

Response: Thank you for the comments. The current study utilizes the linearized ozone module (Linoz) for two parts:
1. Online and interactive ozone calculation. Linoz is the linearized ozone module of the E3SMv2 and feedbacks on the climate, and thus we refer it as "interactive".
2. Steady-state ozone (SSO) diagnosis from offline Linoz. This is a novel use of the SSO metric that helps to separate the chemistry and transport impact of QBO-ozone.

To avoid confusion, we modified the text to refer to Linoz in E3SMv2, while solely refer to SSO when diagnosing chemical-driven ozone. Please refer to line 139-153 and line 364-367.

Need to note that although dynamical nudging is a relatively strong constraint, the ozone impact may still feedback on the climate. This is shown in E3SMv2 Linoz-v2 simulation and fixed ozone simulation (Figs. 4,5, and S8) – interactive ozone decreases QBO-temperature by around $\pm 1$ K, indicating its damping effect on QBO. Please refer to last paragraph of section 6.1 for further detail.

Other comments
1. Please make it clear what you mean by "phase asymmetry" (e.g. line 74, line 179, etc.). Based on further reading in the manuscript and familiarity with the observations, it appears you mean the difference in the length of time and evolution of the wind change with pressure between the easterly and westerly phases. This should be defined at the outset so the reader is not wondering what asymmetry you are referring to.

Response: The "phase asymmetry" refers to asymmetry between QBO phases – a stronger and shorter QBO easterly phase and weaker and longer QBO westerly phase. Please refer to line 57 – 58 for detail.

2. (l. 326-327) Sentence beginning "This indicates …". Since the response of column ozone is mainly driven by the winds in the lower stratosphere, this discrepancy likely indicates that the internally generated QBO is too weak there.

Response: It is revised to "Since the response of column ozone is mainly driven by the wind in the lower stratosphere, this discrepancy likely indicates that the internal generated QBO is too weak there". Please refer to line 331-332.

3. (l. 349-356) What are you trying to say here? The figure indicates poor response of ozone in the simulation with internally generated QBO. What is the "improved representation" that is referred

to?

Response: The improved representation referred to the E3SMv2 Linoz-v3 "$NO_y$-$N_2O$-$CH_4$-$H_2O$ chemistry", which had improved the deficiency in E3SMv2 Linoz-v2 simulation at around 10-hPa. To eliminate the confusion, the "improved representation" sentence is removed. Please refer to line 368-369.

4. Is anything additional learned from the convoluted calculation of the ozone that CESM2 would have if it used Linoz?

Response: The CESM2 steady state ozone calculation reflects the temperature effect on ozone. It is derived as a demonstration of the SSO on non-interactive ozone chemistry. It may also be an indication of the "would-be" temperature-ozone if Linoz were to be implemented in CESM2. To avoid confusion, we've revised the paper and remove the latter expression. Please refer to line 522.

Editorial comments
1. (l. 394) "the no response" -> "the lack of response"
2. (l. 427) What does "colder/warmer" mean? It's either one or the other.
3. (l. 469) "with the with the"
Response: Revised as suggested.

**Editor comments on manuscript egusphere-2024-1927 (revised version)**

We thank editor for making comprehensive comments. The revision are made accordingly. The response to the comments are as follows, the lines are referring to the trackchange version.

**Specific comments:**

P1, L22: It is not clear what is meant with "modeled changes temperature". Please rephrase.

P1, L25: For me it is not clear if these are detected by these changes or if these are caused by these changes.

P2, L35: Add "tropical" before "stratosphere" since this relates to the tropical stratosphere not the entire stratosphere.

P2, L38: Please rephrase the sentence. The QBO itself does not influence the chemical processes in the atmosphere rather the changed composition of the atmosphere due to the QBO is affecting chemistry.

P2, L60: "Nudging of the winds is unphysical and produces an anomalous BDC….". Is this really true. I had a quick look at the Orbe et al. paper and to my understanding they just stat that care should be taken then SD simulations are compared.

Response: The comments are revised as suggested.

P3, L65ff: To my knowledge Linoz is providing only very basic, simplified chemistry which I guess is sufficient for climatic studies. Nevertheless, I would appreciate a statement about the general quality/accuracy of the Linoz scheme.

Response: We thank for the comment. Studies have verification the validity of Linoz in representing the ozone chemistry in stratosphere (Mclinden et al., 2000), and also shown ozone response to 4xCO2 (Meraner 2020). We've added these in the text. Please refer to …..

P3, L83 and 84 and throughout the manuscript: As already stated in my comments before publication in the discussion you should use the Copernicus style and abbreviate write Section beginning with a capital letter and abbreviate it as "Sect." unless it appears at the begin of the sentence.

Response: Revised as suggested. All sections are abbreviated as "Sect."

P4, L121: Were tuned ? Please specify this. What exactly has been done?

P5, L135: A reanalysis is no observation even if observations have been assimilated!

Response: Revised as suggested.

P7, L186: "QBOw": I guess you mean here the QBO west. This abbreviation should be introduced.

Response: Thank you for the comment. The QBOw has been previously introduced. Please refer to line 335.

P8, L220: Please use here a clear writing for dates. Do you mean September 1970 to March 1972?

Response: 1970 September to 1972 March.

P9, L255: Please use the Copernicus style. Equation should be abbreviated as Eq. in the text.

Response: Revised as suggested.

P9, L256: Has the abbreviation SSO been introduced?

Response: Revised as suggested. Please refer to line 87.

P9, L266: "The center" not clear. Do you mean the center of the composite is where the NPLA shift is? Please rephrase so that it becomes clear where the center is.

Response: Yes. The center refers to the "center of the composite". It is revised in the text, please refer to line 428.

P9, L274-275: Rephrase/Improve sentence.

Response: Revised as suggested.

P10, L287-291: Repetition of what has been said before???

Response: It refers to the temperature pattern instead of wind pattern referred before. We revised to avoid confusion. Please refer to line 449-453.

P11, L318: "The TCO patterns exhibits tri-pole pattern of anomalous low in the tropics and high in the extratropics during …." Sentence not clear since something is missing here. Please correct.

Response: It referred to the reanalysis TCO, we revised as suggested. Please refer to line 503.

P11, L320: "The magnitude of the negative in QBOe"? Not clear. Please correct/rephrase.

Response: It is changed to "The magnitude of the low TCO in QBOe." Please refer to line 506.

P11, L335: Has the abbreviation "CMZM" been introduced?

Response: It is the Concentration Monthly Zonal Mean (CMZM) product (Sofieva et al., 2023) mentioned in previous paragraph, please refer to line 291.

P12, L341: "low" -> do you mean "low ozone"? Please clearly write this.

P12, L344: Change sentence as follows: Since the nudged E3SMv2 simulation uses the Linoz-v2 scheme where the chemical species such as ….

P12, L347: Change to "with chemistry of NO2-N2O-CH4-H2O included".

Response: Revised as suggested.

P12, L348: Missing parts of what?

Response: Missing ozone fluctuation between 6-hPa to 10-hPa. Please refer to line 536.

P12, L366: Correct as follows: " ……phase changes in the northern hemisphere."

Response: Revised as suggested.

P14, L398: What is noisier? The temperature? Please clearly state this.

Response: The temperature is noisier. Revised as suggested.

P14, L403: Either singular or plural, thus either "like stratospheric sudden warmings" or "like a stratospheric sudden warming". Further "stratospheric" and "sudden" need to be swapped. It should read "sudden stratospheric warming".

Response: Revised as suggested.

P14, L418: "To further examine the variable responsible for the change, the single specie sensitivity….." Please improve sentence.

Response: Revised as suggested.

P14, L426: Please clearly state which level.

Response: Revised as suggested.

P15, L431-432: Check sentence and rephrase. Add "that is" before controlled? What is meant with "return arm".

Response: It is a return branch of the QBO-induced circulation. Revised in the text. Please refer to 705.

P16, L441: Similar than what? Please use clear statements.

Response: Similar temperature and $\underline{w}^*$ patterns. It is revised as suggested. Please refer to 714-715.

P15, L442: Change to "…. Similar prognostic ozone if Linoz were to be implemented….."

Response: Revised as suggested. Please refer to line 715-716.

P14 and P15: References to the figures are given in a very confusing order. You are jumping here back and forth between the figures which makes it very hard to follow.

Response: Revised as suggested. We revised the figures to avoid jumping back and forth.

P15, L455: Here references should be given. I assume this statement is based on results from previous studies.

Response: The statement is based on the evaluation of the metrics for the current panel Figure 13a.

P16, L459ff: NH -> northern hemisphere, SH -> southern hemisphere. Please use a consistent writing.
P16, L460: Also here references should be added.

Response: Revised as suggested.

P16, L487: "less field of high wavenumbers" not clear

Response: Revised as suggested. It should be "less spectrum of zonal wavenumbers". Please refer to line 742.

P17, 489: "with nudging" -> please rephrase
P17, L502: Check sentence and please correct.
Supplement: Check numbering of figures. Figure S7 appears twice.

Response: Corrected as suggested.

**Figure captions in general:**
These should start with "Figure" and the respective number followed by a colon. "Figure" is only abbreviated in the text!
anomalous -> anomaly
use a readable date formats.
Add degree sign to the given coordinates.
Figure 8 caption: OBS? Do you mean "observations"?
**Technical corrections:**
P2, L33: add "atmospheric" before "circulation".
P2, L33: that -> and
P2,L55:  Write "zero"  instead  of "0".
P4, L104: about -> of (?)
P6, L179: QBOs -> QBO
P10, L287: tropics -> tropical
P10, L302: add "the" -> the northern
P10, L303: accordance? Do you mean "agreement"?
P10, L305: extratropics -> extratropical
P11, L307: is -> are
P11, L310: creates more -> creates a more
P11, L334: extratropics -> extratropical
P12, L345: chemistry -> chemical
P12, L354: add "being" so that it reads "being weaker".
P12, L355: add "the" -> the E3SM-v2
P13, L367: hemisphere -> hemispheres
P13, L375: section 4 -> Sect. 4

P13, L374: equation 2 and 3 -> Eq. 2 and 3
P13, L389 and 392: correspond -> corresponds.
P14, L398: Southern Hemisphere -> southern hemisphere
P15, L439: process -> processes
P16, L483: Add "the" so that it reads "by the two models".
P16, L439: change -> changes
P16, L462: write "and" instead of using a comma.
P16, L465: Conclusion -> conclusion
P16, L467: Firstly -> First
P16, L468: dynamic -> dynamical
P16, L468: add "the" so that it reads "of the QBO".
P16, L469: delete "while the".
P16, L470: profile -> profiles (?)
P16, L 471: Delete "the"
P16, L475: difference -> different
P16, L482: degree sign missing.
P17, L488: Lastly -> Last
P17, L507 and 509: Add "the" so that it reads "the nudged" and "the QBO", respectively.
P17, L512: process -> processes
P17, L515: Delete "change".
Response: We thank the editor's thorough comments. These are revised as suggested.

---

## Author Response (AR3)

Item-by-item response to all review comments

NOTE: To facilitate the evaluation of our responses, original review comments are listed first in their originals (in black), followed by our itemized response (in blue). An annotated version of revised manuscript is attached.

We thank the reviewers' comments, which are in black text below. Our response is followed (in blue).
* * *
Reviewer 1

I am pleased to inform you that your manuscript is accepted for publication after the following corrections have been considered:

P5, L131: "can" or "to". One of the words is obsolete.

P6, L161: El Nino and Southern Oscillation -> El Nino Southern Oscillation

P8, L219: Remove full stop after years (1970 and 1972)

P8, L242: You mean "as" in McLinden et al.? Has the same term set to zero in McLinden et al. or are only you doing this?

Response: Revised as suggested.

Section 3 and 4: Aren't these belonging to the method section? I would suggest to add these to section 2 as subsections.

Response: We've moved these sections to section 2 and modified accordingly.

P11, L307: What do you mean with phase change? Do you mean shifted?

Response: It is phase shift, revised as suggested.

P11, L326: What is meant with natural? Do you mean the reference run?

Response: This means the E3SMv2 natural run which is done by making a free-run with no nudging. We've made the modifications in the text. Please refer to line 165.

P12, L348: which -> where?

Response: Revised as suggested.

P15, L433: What do you mean with return branch? The downward branch of the BDC? If yes, then please write it like this.

P15, L434: What is meant with phase reversal in the tropics. Not clear, please rephrase sentence.

Figures 4-11: These figures are of similar type and in the caption always the same text is repeated. I think it would be easier if you have a detailed description in the first of these figures and then write for the following figures "As in Fig. X, but for........." and just point out what the differences are.

Response: Revised as suggested.

Reviewer 2

The authors have made major revisions to the manuscript in response to my previous comments. I apologize that I was not able to follow the motivation or method of the earlier versions. The latest revision is now straightforward to follow and I hope will be so for other readers. The substitutions of the terms "metric" and "SSO" for "model" to describe the steady-state solution was very helpful for keeping various options for ozone straight. I recommend that the paper be accepted after attention to a few minor comments given here.

1. I sometimes was confused about which simulation of E3SMv2 was being referred to. On some occasions the terms "E3SMv2 nudged", "E3SMv2 natural" and "E3SMv2 Linoz-v3 nudged" are used whereas elsewhere just "E3SMv2" is used. It would be helpful for the reader if the names for the model runs were consistently used throughout.

Response: We've added the naming reference to the E3SMv2 nudged and natural simulations in the text to make further clarification of the simulations. Please refer to line 161 and 165.

2. As a related matter, I thought the natural run was the one with QBO generated by parameterized gravity waves, which has a weak QBO with a period of 27 months. Then I saw this (line 355-6): "the period of internally generated QBO in E3SMv2 is ~21 years". Please clarify.

Response: There was a typo in the sentence. The Tuning is referred to changes in QBO period for EAMv1. Revised in the text. Please refer to line 100-102.